# At-Home Orthodontic Treatment for Severe Teeth Arch Malalignment and Severe Obstructive Sleep Apnea Syndrome in a Child with Cerebral Palsy

**DOI:** 10.3390/ijerph19095333

**Published:** 2022-04-27

**Authors:** Atsuko Tamura, Kohei Yamaguchi, Ryosuke Yanagida, Rie Miyata, Haruka Tohara

**Affiliations:** 1Department of Dysphagia Rehabilitation, Graduate School of Medical and Dental Sciences, Tokyo Medical and Dental University, Tokyo 113-8510, Japan; tamura.d.c@nifty.com (A.T.); ry.yanagida@gmail.com (R.Y.); harukatohara@hotmail.com (H.T.); 2Department of Pediatrics, Tokyo Kita-Medical Center, Tokyo 115-0053, Japan; rie88miyata@yahoo.co.jp

**Keywords:** cerebral palsy, dysphagia, respiratory function, orthodontic treatment, obstructive sleep apnea

## Abstract

Children with cerebral palsy typically have severe teeth arch malalignment, causing swallowing and respiration dysfunction. Malalignment in cerebral palsy, especially in children, worsens dysphagia and respiratory disorders; sometimes, it is also noted with obstructive sleep apnea. However, no study has reported on the improvement in obstructive sleep apnea after at-home orthodontic treatment in children with cerebral palsy. We herein present a pediatric case of cerebral palsy wherein obstructive sleep apnea improved with at-home orthodontic treatment for malalignment. We administered at-home orthodontic treatment to a 15-year-old boy with quadriplegia, due to spastic-type cerebral palsy, having no oral intake, obstructive sleep apnea, and teeth arch malalignment. After treatment, a decline in the severity of sleep apnea was observed. Perioral muscle hypertension and oral intake difficulties cause maxillary protrusion, narrowed teeth arch, and tilting of teeth in children with cerebral palsy. We expanded the oral cavity volume by orthodontic treatment to relieve muscle hypertension and correct the tongue position, thereby remarkably improving obstructive sleep apnea. Our findings suggest that at-home orthodontic treatment for malalignment effectively improves perioral muscle hypertension, glossoptosis, and obstructive sleep apnea.

## 1. Introduction

Cerebral palsy (CP) is a syndrome caused by brain damage and is characterized by non-progressive neuromuscular disorders. Movement disorders and palsy from the fetal or infancy periods are its clinical symptoms, which depend on the damaged area, extent, and severity of the brain lesions. CP is a collection of disturbances of motor and posture development and causes restriction of activities, due to non-progressive disorders of the brain during fetal and infancy periods. The disturbance of motor development includes sensation, cognition, communication, recognition, activity, and paroxysmal disease. An international workshop in the USA tried to update the definition of CP [1], and it is said that developmental disturbance due to underweight birth should be comprehensive in a diagnosis of CP [2]. Prenatal risk factors of CP are (1) premature birth (<36 weeks), (2) underweight birth (less than, or equal to, 2500 g), (3) intrauterine infection, (4) multiple births, and (5) uteroplacental insufficiency. Perinatal risk factors are (1) neonatal asphyxia, (2) caesarian section, (3) hyper- or hypo-glycemia, (4) periventricular leukomalacia, (5) intracerebral hemorrhage, and (6) cerebral hemorrhage. Postnatal risk factors are (1) infection, (2) epilepsy, and (3) hyperbilirubinemia [3,4,5,6,7,8,9,10,11,12,13,14]. The occurrence of CP is around 1.5 to 2 people per 1000 people. It is reported that 70–80% of cases of CP are due to causes in the fetal period, but some studies have reported cases in the perinatal or postnatal period [15,16]. The occurrence of CP is related to body weight at birth: low birth weight is associated with periventricular leukomalacia, a type of hypoxic-ischemic encephalopathy which leads to spastic diplegia. Additionally, CP could be caused by the aftereffects of brain damage caused by postnatal diseases, including meningitis, injury, status epilepticus, and kernicterus caused by the progress of hyperbilirubinemia. Hagberg et al. revealed the most common timepoint of CP diagnosis in a long-term study in the West Wedel region of Germany, using findings from computed tomography and magnetic resonance imaging [17,18,19,20,21].

CP is classified according to motor dysfunction, the extent of dysfunction, and severity. Motor dysfunction is classified as spastic type (70% of CP), athetoid type (involuntary movement type), mixed type, palsy type, and hypotonic type. The extent of dysfunction is classified as tetraplegia, diplegia, and hemiplegia. Severity is classified using the Gross Motor Function Classification System (GMFCS), which can predict the prognosis of CP [22,23]. Children with CP usually have many complications [24]. Hearing impairment (sensorineural deafness), visual impairment (nystagmus, squint, atrophy of optic nerve, ametropia, uveitis), language disorder (spoken language), intellectual disability (verbal IQ, performance IQ), perceptual disorder, and epilepsy are often seen in children with CP [25]. Hip dislocation, chronic pulmonary disease, scoliosis, and cervical spondylotic myelopathy are secondary disabilities, accompanied by contracture and dysphagia with aging. Scoliosis is caused by asymmetric posture, due to hip dislocation and unstable seating as a result of abnormal muscle tonus. In addition, spine malformation causes thorax malformation, which leads to malformations in the bronchi and lungs, causing respiratory disturbances, including restrictive ventilation impairment, respiratory distress syndrome (common among premature babies), apnea attack, and the disturbance of coordinated movement of respiratory muscles.

More than half of children with CP have malocclusion in the oral cavity, and most of them are classified as Angle Class II. The symptoms in the oral cavity are teeth attrition caused by bruxism and clenching, open bite, large overjet, hypotonicity of perioral muscles, tongue protrusion, and malalignment of narrowed teeth arch caused by mouth breathing. Bruxism is a significant finding, found in 36.9% of children with CP, and causes occlusion wounds of the lip and tongue [26,27,28]. Malalignment of the teeth arches is caused by involuntary and hypertonic facial, masticatory, and tongue muscle movement, and its severity depends on the level of brain damage [29,30,31]. Malalignment in children with CP exacerbates dysphagia and respiratory disorders and is sometimes diagnosed with obstructive sleep apnea (OSA). Although malalignment is one of the factors for OSA [32,33,34], no study has reported OSA improvement following at-home orthodontic treatment in children with CP. We herein report a pediatric case of CP wherein OSA improved following at-home orthodontic treatment for teeth arch malalignment.

## 2. Case Presentation

The patient was a 15-year-old boy with poor head control because of weakness, thoracic deformity, and scoliosis; the patient’s height and weight were 163 cm and 27 kg, respectively. The patient was born by vaginal delivery at a gestational age of 39 weeks and 5 days, with a birth weight of 3395 g. The patient received breast milk 7–8 times per day. Rollover while sleeping, pursuit, and head control appeared at 3 months old, but the patient developed afebrile convulsion status, acute subdural hematoma, and cerebral atrophy at 4 months old. From 6 months of age, epileptic spasms appeared.

Consequently, the patient underwent hospitalization with his mother for 1 year. During hospitalization, the patient received breast milk by bottle-feeding. However, he started to use nasogastric tube feeding, due to recurring aspiration pneumonia. He started oral food intake at the age of 2 and started to eat by himself at the age of 4. At that time, he used tube feeding, except for lunch with paste food, and semi-caregiving oral intake. He used to finish lunch during elementary school with a strong appetite. However, due to pneumonia recurring every 3 months, he underwent gastrostomy at the age of 10. He stopped oral intake at the age of 13 because of silent aspiration, which was detected by sudden decrease of saturation during the meal. After stopping oral intake, he no longer experienced pneumonia. He was referred to our clinic at the age of 15 by a medical doctor. It was difficult for him to care for his mouth by himself; hence, his mother mainly cared for it. He underwent annual or biannual dental check-ups at a daycare facility, and no obvious caries or gingivitis were found. He was significantly emaciated and had appendicular contracture; hence, it was impossible for him to maintain his posture on the dental treatment chair. Considering the need for his visits to a dental clinic, we provided at-home dental treatment. We carried out the treatment on a mat with the patient in the supine position. 

At 15 years of age, the upper and lower deciduous canines and second deciduous molar remained, the lower first molar was semi-impacted, and the remaining teeth had erupted randomly but not on the alveolar ridge. The tongue was rolled posteriorly in the oral cavity, due to hypotonicity with significant glossoptosis. A narrowed maxillary teeth arch, high-arched palate, significant tilting mandibular anterior teeth, and teeth arch malalignment were observed in the oral cavity, causing a significant decrease in oral cavity volume (Figure 1). 

The patient was diagnosed with OSA due to frequent apnea and declining oxygen saturation at night; the apnea-hypopnea index (AHI) and minimum saturation were 32.3 and 82%, respectively, and biphasic positive airway pressure was introduced while sleeping. However, the symptoms did not improve due to poor compliance. He tended to wake up every 2 or 3 h because of discomfort due to the ready-made mask. The patient’s medical doctor contacted us regarding the oral problems. Orthodontic treatment was subsequently initiated considering the severe malalignment of the teeth and severity of OSA as indicated by the AHI, and in consultation with the patient’s medical doctor, regarding the necessity of oral treatment.

As we considered that malalignment of the mandibular teeth arch was associated with glossoptosis, we started orthodontic treatment using a removable appliance to the tilted lower anterior teeth on the lingual side. The frequency of treatment was once a month or every 2 months. The patient’s compliance was not satisfactory with the use of the removable appliance and setting and removing the appliance imposed a burden on the caregiver. After 2 months, a fixed multi-bracket appliance was used instead. After 7 months, the crowding of lower anterior teeth had almost improved. An additional fixed multi-bracket appliance was introduced to improve the narrowed maxillary arch. The length and width of the upper arch decreased by 4 mm and 2 mm, respectively, and those of the lower arch increased by 9 mm and 7 mm, respectively. As a result of these changes, the upper and lower arches became well-balanced, and the volume of the oral cavity improved well after our intervention (Figure 2). Although the fixed multi-bracket appliance detached several times, the soft tissues were not injured by the wire during the orthodontic treatment. The oral malfunctions, due to the malalignment of teeth arches, including interference in oral lip closure and retraction of lingual position, improved after orthodontic treatment. It became possible for the patient to protrude the tongue to the lower lip. The dryness of the mouth was relieved, the oral hygiene status was favorable, and new dental caries did not appear during the treatment period, owing to the support of the patient’s family. Saliva pooling was significant at first, but the amount of saliva retention decreased as the treatment progressed. The patient underwent an overnight polysomnography examination before and after the orthodontic treatment. At first, the AHI was 32.3, and the minimum saturation was 82%. The severity of both central and OSA syndromes was high. Significant improvement was observed after our intervention. After the orthodontic treatment, the AHI was 6.0, and the minimum saturation was 90%. The severity of obstructive sleep apnea syndrome (SAS) was low, whereas that of central SAS was high. In 2016 (at 17 years of age), the AHI worsened due to brain-wave deterioration. Following this, the convulsive seizures were controlled by medication, and the AHI declined from one-sixth to one-eighth of the pre-intervention AHI (Figure 3). A fixed multi-bracket appliance is currently attached to the labial surface. We plan to bond a wire to the palatal and lingual surfaces for orthodontic retention in the future.

## 3. Discussion 

CP is referred to as an “umbrella term” because CP is not a single disease and can occur due to various reasons at various time points [35]. Due to the characteristics of its symptoms, CP is associated with many complications, including permanent motor and posture disorders; hypertension of the buccal mucosa, oral cavity, and tongue; and the malalignment of teeth arches, including a narrowed teeth arch, high-arched palate, and maxillary protrusion [26,28]. Therefore, children with CP frequently develop OSA [33,34,35], although the percentage of undiagnosed OSA is high. OSA causes obstruction of the pharynx and airway due to the imbalance between the pharyngeal anatomical structure of bones, such as the maxilla and mandible. and soft tissues, such as the tongue and soft palate. Consequently, dentistry plays an important role in sleep medicine. The child in this report had OSA caused by upper airway obstruction, due to the significant malalignment of teeth arches and glossoptosis [36].

Continuous positive airway pressure (CPAP) and oral appliances are widely spread symptomatic treatments. Although the effectiveness of CPAP is high, patient compliance often becomes poor [36,37]. Oral appliances are also widely used [19], which are less effective, but are associated with higher convenience and compliance [20]. However, compliance with oral appliances is broad, ranging from 0% to 70% [38,39]. In this case, the patient started to use CPAP at the age of 14 but did not use it due to misfitting of the mask. Additionally, the patient did not have a large enough jaw opening distance to take impressions of the teeth arch with a ready-made dental impression tray, and it was unclear whether the patient could continuously wear an oral appliance while sleeping.

Sleep surgery and orthodontic treatment are curative treatments for OSA, and they can improve ventilation obstruction in the upper airway. Additionally, myofunctional therapy (MFT), which improves discrepancies of the perioral muscles, such as the resting tongue position and lip position, is considered an adjuvant therapy for OSA [40,41]. Certain treatment effectiveness is expected for orthodontic treatment to expand the oral cavity after improving malalignment of teeth, regardless of patient compliance. In cases with spastic-type CP, many patients have difficulty opening the jaw, due to the hypertension of the perioral muscles. The patient in the present case had no difficulty in opening his mouth and did not experience any spastic movement of the body; hence, dental treatment was relatively easy [42,43]. Therefore, we selected orthodontic treatment to improve the significant malalignment to expand the volume of the oral cavity to improve OSA in this case. Following our at-home orthodontic treatment to expand the teeth arches, a significant improvement was observed in both AHI (from 33.0 to 6.04) and minimum saturation (from 82% to 90%). We considered that both the length and width of the upper and lower teeth arches improved, the volume of the oral cavity increased, and the OSA improved (Figure 2). Patient compliance is also an important factor when deciding on a treatment plan for OSA.

In this case, we started at-home dental treatment because of the patient’s difficulties in walking and instability due to significant scoliosis. In Japan, where 29.1% of the population is considered elderly according to a report from the Statistics Bureau of the Ministry of Internal Affairs and Communications, at-home dental treatment is frequently practiced. To improve the systemic function related to oral and swallowing function for physically non-dependent patients, at-home dental treatment is extremely effective. However, it is difficult to conduct the same examinations at someone’s house that one would perform in an outpatient department at a hospital or clinic. Problems encountered include shortage of dental equipment, difficulties in vapor proofing, and insufficient light sources. A definitive diagnosis for OSA is usually made with polysomnography examination. In this case, the patient underwent polysomnography examination during his regular semi-annual hospitalization and received the diagnosis of OSA. Dental cephalograms, which visualize the upper airway, are effective for the comprehensive evaluation of the maxillofacial region and upper airway, but they are impossible to conduct at a patient’s home. The treatment purpose, in this case, was to increase the volume of the oral cavity rather than improve teeth arch alignment. Therefore, we evaluated the effectiveness of treatment using sleep- and respiratory-related factors such as AHI. However, while the balance of the maxillofacial region and the location of the hyoid bone can be confirmed by a cephalogram, the evaluation of soft tissues is difficult because children with CP tend to have muscle hypertonicity when standing or sitting [44,45,46,47,48]. Pediatric patients with CP usually have hypertension. One of the reasons that the patient in this case experienced successful treatment is that at-home treatment was more familiar to the patient, compared to a dental chair at an outpatient department.

Our orthodontic treatment lasted six years. The reasons for this duration were the weak orthodontic force and the suspension of treatment due to the patient’s hospitalization and the COVID-19 pandemic of. The patient may have felt uncomfortable during the treatment because of the fixed multi-bracket appliance. In addition, the appliance increased the risk for gingivitis and dental caries. In this case, we provided a regular dental checkup and tooth brushing instructions for the mother to reduce the risk of these possible adverse effects. We considered that preventing teeth arch malalignment is important as at-home orthodontic treatment for children with cerebral palsy burdens both dentists and caregivers. 

According to the polysomnography findings of this case, both obstructive and central SAS were significantly improved through the administration of perampanel. Both central and obstructive symptoms are present in SAS in children with CP; therefore, both medical and dental approaches are effective. A decrease in the capacity of the oral cavity causes glossoptosis, leading to not only OSA but also systemic adverse effects, including digestive problems [42]. Caregivers should be aware of systemic problems associated with oral issues to identify the effect of malalignment on respiratory disorders, dysphagia, and malnutrition.

## 4. Conclusions

Children with OSA sometimes have severe malalignment of the teeth arch, which leads to a decrease in volume of the oral cavity. This study revealed that at-home orthodontic treatment to expand the teeth arch is an effective way to improve symptoms of OSA.

## Figures and Tables

**Figure 1 ijerph-19-05333-f001:**
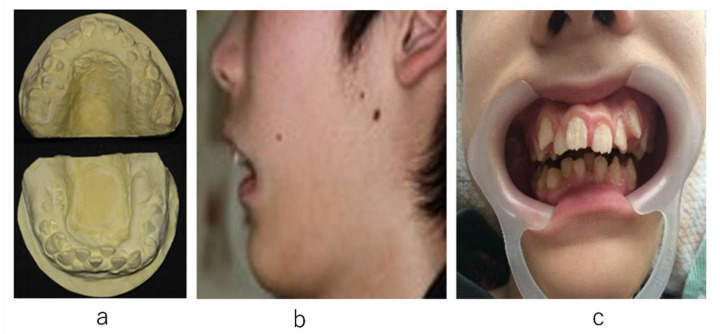
Initiation of orthodontic treatment. (**a**): Study model. (**b**): Facial profile. (**c**): Frontal view of the dental arch at the first visit.

**Figure 2 ijerph-19-05333-f002:**
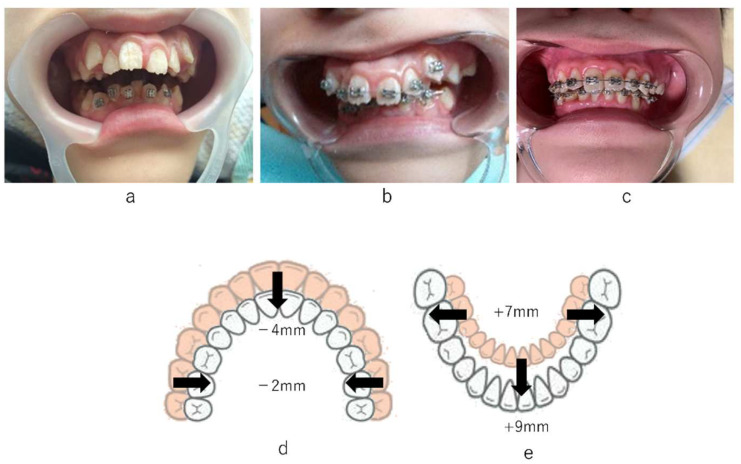
Prognosis of orthodontic treatment. (**a**): Fixed multi-bracket appliance placed on the mandibular anterior teeth (2 months after starting orthodontic treatment). (**b**): Fixed multi-bracket appliance placed on the maxillary anterior teeth (7 months after starting orthodontic treatment). (**c**): View of the anterior teeth at 6 years after the initiation of orthodontic treatment. (**d**): The length and width of the upper arch decreased by 4 mm and 2 mm, respectively. (**e**): The length and width of the lower arch increased by 9 mm and 7 mm, respectively.

**Figure 3 ijerph-19-05333-f003:**
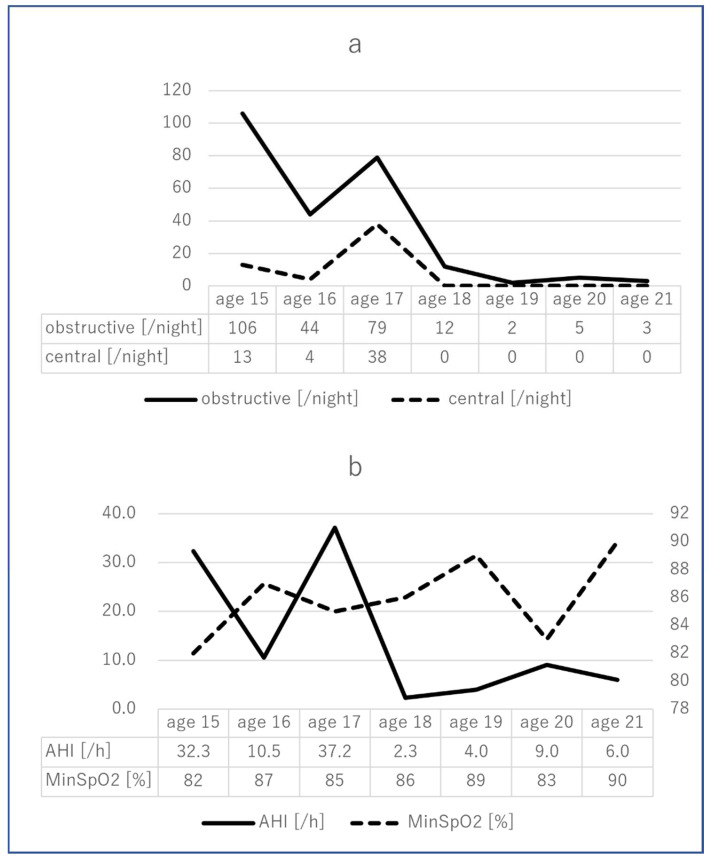
Time course of changes in the polysomnography examination. (**a**): The total number of obstructive and central sleep apnea events during sleeping at night. (**b**): Apnea–hypopnea index (AHI) score and minimum oxygen saturation (SpO_2_) during sleeping at night.

## Data Availability

The datasets used and/or analyzed in the current study are available from the corresponding author on reasonable request.

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
