# Peer review of "At-Home Orthodontic Treatment for Severe Teeth Arch Malalignment and Severe Obstructive Sleep Apnea Syndrome in a Child with Cerebral Palsy"

_ijerph, 2022, doi:10.3390/ijerph19095333_

Round 1

Reviewer 1 Report

This case report covers the topic of : At-Home Orthodontic Treatment for Severe Teeth Arch 2 Malalignment and Severe Obstructive Sleep Apnea Syndrome 3 in a Child with Cerebral Palsy. The manuscript is well written and covers a very interesting topic and could be very helpful for conduction future clinical trials on this project. The introduction and the discussion are very thorough. The reviewer recommend more details in the treatment sequence such as how hard or easy was to insert the fixed appliance. What technique was used to manage this patient. This would benefit the reader if added. 

Author Response

Point 1: Malalignment and Severe Obstructive Sleep Apnea Syndrome 3 in a Child with Cerebral Palsy. The manuscript is well written and covers a very interesting topic and could be very helpful for conduction future clinical trials on this project. The introduction and the discussion are very thorough. The reviewer recommend more details in the treatment sequence such as how hard or easy was to insert the fixed appliance. What technique was used to manage this patient. This would benefit the reader if added. 

Response 1: Thank you for this observation. We have added the patient‘s background and treatment details to the revised manuscript.

Reviewer 2 Report

  1. The paper requires appropriate paragraphing throughout, as there are huge chunks of text, which could be overwhelming for the reader; please do this to enhance the presentation.
  2. Please ensure that the single Discussions and Conclusions section is separated into distinct, separate Discussion and Conclusion sections.
  3. Is the sentence on line 96: “He finishes elementary school lunch with a strong appetite” meant to be: “He used to finish elementary school with a strong appetite” ?”
  4. The sentence on lines 110-111: “Orthodontic treatment was started according to a strong request by the patient’s family.” Should treatment be started just because a patient’s family request it? Why else were you compelled to commence such treatment? What other factors were taken into account? Did the benefits outweigh any of the risks?
  5. In lines 180-181, you mention: “However, the patient in this case always opens his mouth, and so we could perform orthodontic treatment.” This sounds as though just because someone can open their mouth, they are able to have orthodontic treatment. What other factors were taken into account to ensure that the patient was a realistic candidate for orthodontic treatment?
  6. What was the patient’s relevant dental history? Were they a regular dental attender? How was their oral hygiene? Had they ever had any type of dental treatment (i.e., scaling, polishing, caries removal and restorations)? Was the patient able to maintain optimal levels of oral hygiene themselves, or did they receive assistance from caregivers?
  7. As per line lines 144-145: “View of the frontal teeth at 6 years after the initiation of orthodontic treatment” along with the corresponding photograph (c), did the at-home orthodontic treatment take 6-years to complete?
  8. As per lines 170-171: “the patient did not have a large enough jaw-opening distance to take impressions of the teeth arch with a ready-made dental impression tray,” how were dental impressions taken, or were any alternate methods used?
  9. In lines 187-188, you state: “Patient compliance is also an important factor when deciding a treatment plan for OSA.” However, there is very little description about the compliance of the patient within the case. Were they compliant? Did their compliance grow? How did they cope with the orthodontic treatment? How was their compliance different from an adolescent without such special needs? How often was this patient seen for the orthodontic treatment? What was the total number of appointments? Were there any orthodontic emergencies?
  10. Lines 189-206 discusses a few minimal reasons why “at-home” treatment was performed for this patient. This treatment modality needs to be considered in more detail, since it is more unique compared to in-surgery treatment. What were the benefits of at-home treatment for the patient, caregivers, and dental team? What were the limitations? Who attended the patient’s home (i.e., clinician, dental assistant)? How was the patient treated in the at-home environment (i.e., on a chair, lying down in bed)? The at-home setup and providing at-home orthodontic care needs much further discussion in order to inform the reader.
  11. Lines 199-201 state that “Dental cephalograms, which visualize the upper airway, are effective for the comprehensive evaluation of the maxillofacial region and upper airway, but they are impossible to conduct at a patient’s home.” Did the patient receive any dental radiography before orthodontic treatment commenced? If not, why was this the case and was this appropriate? If another type of dental radiograph was taken, why was this, and can this be included within the manuscript?
  12. The paper has a great focus on: “we selected orthodontic treatment to improve the significant malalignment to expand the volume of the oral cavity to improve OSA in this case,” as stated in lines 181-182. However, what about actual follow-ups and maintenance after the orthodontic treatment? Is the patient seeing a dentist regularly for oral health maintenance? What was the prognosis of the orthodontic treatment? What about retention?
  13. Are there any further clinical photographs that can be provided, which further show the progress and transition of treatment?
  14. A recent publication (Modha, B., 2021. Global Developmental Delay and Its Considerations in Paediatric Dental Care—A Case Report. Oral1(3), pp.181-189) contains similar material to your Introductory material, particularly lines 75-76: “Malalignment of the teeth arches is caused by involuntary and hypertonic facial, masticatory, and tongue muscle movement, and its severity depends on the level of brain damage;” this publication would be a useful one to reference here.
  15. Some other useful and helpful reads and references for your paper: (Abeleira, M.T., Outumuro, M., Diniz, M., García-Caballero, L., Diz, P. and Limeres, J., 2016. Orthodontic treatment in children with cerebral palsy. Cerebral Palsy-Current Steps, pp.130-40), (Napoli, J.A., Drew, S. and Jaeger, T.C., 2020. Management of Skeletal Facial Deformation and Malocclusion in Cerebral Palsy. Cerebral Palsy, pp.1105-1120)
  16. Generally, this paper is quite easy to read. The figures and references are fine. Yet, please kindly note, the language, grammar, punctuation, spelling and sentence structures within the paper, all must be carefully assessed and refined to ensure a succinct and coherent read. This is because there are minor discrepancies in the language, grammar, punctuation, spelling and sentence structures. Please obtain the necessary scientific English language reading and editing assistance, if need be, so that the paper has the potential to be read enjoyably by the international readership.
  17. Well done for providing this patient with life-changing orthodontic treatment, which I hope has improved their quality of life.

Author Response

Point 1: The paper requires appropriate paragraphing throughout, as there are huge chunks of text, which could be overwhelming for the reader; please do this to enhance the presentation.

Response 1: Thank you for this valuable comment. We have divided the huge chunks of text into smaller paragraphs to enhance readability.

Point 2: Please ensure that the single Discussions and Conclusions section is separated into distinct, separate Discussion and Conclusion sections.

Response 2: Thank you for this recommendation. We have separated the Discussion and Conclusion into different sections (Page 5, Line 173; Page 6, Line 244).

Point 3: Is the sentence on line 96: “He finishes elementary school lunch with a strong appetite” meant to be: “He used to finish elementary school with a strong appetite”?

Response 3: Thank you for this observation. We apologize for our poor English. We have modified the sentence for improved clarity (Page 3, Lines 98).

Point 4: The sentence on lines 110-111: “Orthodontic treatment was started according to a strong request by the patient’s family.” Should treatment be started just because a patient’s family request it? Why else were you compelled to commence such treatment? What other factors were taken into account? Did the benefits outweigh any of the risks?

Response 4: Thank you for this observation. The patient’s parent experienced difficulties providing oral care because of the patient’s narrowed teeth arch. However, we started our intervention for these three main reasons: (1) severe malalignment of the teeth, (2) severe obstructive sleep apnea with an AHI of 33.0, and (3) consultation from a medical doctor regarding the patient’s oral condition. We have added these reasons in the main text (Page 3, Lines 120–125).

Point 5: In lines 180-181, you mention: “However, the patient in this case always opens his mouth, and so we could perform orthodontic treatment.” This sounds as though just because someone can open their mouth, they are able to have orthodontic treatment. What other factors were taken into account to ensure that the patient was a realistic candidate for orthodontic treatment?

Response 5: Thank you very much for this observation. We have modified the main text in the revised manuscript to clarify the considerations for initiating orthodontic treatment based on this comment (Page 5, Line 202–204).

Point 6: What was the patient’s relevant dental history? Were they a regular dental attender? How was their oral hygiene? Had they ever had any type of dental treatment (i.e., scaling, polishing, caries removal and restorations)? Was the patient able to maintain optimal levels of oral hygiene themselves, or did they receive assistance from caregivers?

Response 6: Thank you for this comment. The patient has no previous history of oral diseases. His daycare center offers annual or biannual dental checks. It is almost impossible for him to ensure oral hygiene independently; hence, his mother usually provided this care (Page 3, Lines 102–106).

Point 7: As per line lines 144-145: “View of the frontal teeth at 6 years after the initiation of orthodontic treatment” along with the corresponding photograph (c), did the at-home orthodontic treatment take 6-years to complete?

Response 7: Thank you for this observation. In our case, the orthodontic treatment took 6 years when we wrote this manuscript. During this period, the patient’s appointments were postponed sometimes because the patient had inspection hospitalization or sudden deconditioning due to the cerebral palsy. Additionally, each treatment time was longer than that in other children. Consequently, we required a longer duration to complete the treatment in this case.

Point 8: As per lines 170-171: “the patient did not have a large enough jaw-opening distance to take impressions of the teeth arch with a ready-made dental impression tray,” how were dental impressions taken, or were any alternate methods used?

Response 8: Thank you very much for this comment. In our case, we installed a removable appliance to tilt the lower anterior teeth to the labial side. The size of the appliance was small (extending between the right and left first premolars); therefore, we could take an impression. However, the appliance's management burdened the patient’s caregiver, and it had an insufficient effect. Hence, we used a fixed appliance. We considered it difficult to take an impression, including the first and second molars (Page 3, Lines 126–131).

Point 9: In lines 187-188, you state: “Patient compliance is also an important factor when deciding a treatment plan for OSA.” However, there is very little description about the compliance of the patient within the case. Were they compliant? Did their compliance grow? How did they cope with the orthodontic treatment? How was their compliance different from an adolescent without such special needs? How often was this patient seen for the orthodontic treatment? What was the total number of appointments? Were there any orthodontic emergencies?

Response 9: Thank you very much for this comment. The reasons for poor compliance with BiPAP were (1) discomfort while using the mask because it was ready-made and (2) sleep deprivation every 2 or 3 hours while using BiPAP. Currently, the patient uses BiPAP for approximately 1 hour once in several days. Hence, he is noncompliant. It is reported that the compliance with OA for OSA is 70%, while that with CPAP is 50%. Generally, compliance with BiPAP and CPAP is worse than that with OA. We carried out orthodontic treatment once a month or every 2 months for a total of 51 times in 6 years. The fixed multi-bracket appliance sometimes disengaged several times; however, the wire did not injure the soft tissues during the course of orthodontic treatment (Page 3, Lines 137–139).

Point 10: Lines 189-206 discusses a few minimal reasons why “at-home” treatment was performed for this patient. This treatment modality needs to be considered in more detail, since it is more unique compared to in-surgery treatment. What were the benefits of at-home treatment for the patient, caregivers, and dental team? What were the limitations? Who attended the patient’s home (i.e., clinician, dental assistant)? How was the patient treated in the at-home environment (i.e., on a chair, lying down in bed)? The at-home setup and providing at-home orthodontic care needs much further discussion in order to inform the reader.

Response 10: Thank you for this comment. The patient was significantly emaciated and had appendicular contracture. Hence, it was difficult for him to walk by himself or maintain his posture on the dental treatment chair. Our home-visiting dentistry allows patients to receive dental treatment in an environment they are accustomed to. During our treatments, two dentists visited at the same time. We carried out the treatment on a mat with the patient in the supine position because the patient is most relaxed in this position (Page 3, Lines 107–109). The disadvantage of home-visiting treatment is the shortage of equipment compared to the outpatient department. Additionally, we experienced difficulties with vapor proofing and light sources (Page 6, Line 216-220).

Point 11: Lines 199-201 state that “Dental cephalograms, which visualize the upper airway, are effective for the comprehensive evaluation of the maxillofacial region and upper airway, but they are impossible to conduct at a patient’s home.” Did the patient receive any dental radiography before orthodontic treatment commenced? If not, why was this the case and was this appropriate? If another type of dental radiograph was taken, why was this, and can this be included within the manuscript?

Response 11: Thank you for pointing it out. In this study, we did not take a radiograph. The malalignment was visually significant, and the tilting mandibular teeth significantly decreased the oral cavity volume. We chose orthodontic treatment because we considered that increasing the volume of the oral cavity increases the high AHI effectively. Our treatment aimed to increase the oral cavity volume, not improve the teeth alignment. Hence, we evaluated the appropriateness of volume increase accompanied with the teeth arch enlargement based on the improvement in the AHI (Page 6, Lines 225–228).

Point 12: The paper has a great focus on: “we selected orthodontic treatment to improve the significant malalignment to expand the volume of the oral cavity to improve OSA in this case,” as stated in lines 181-182. However, what about actual follow-ups and maintenance after the orthodontic treatment? Is the patient seeing a dentist regularly for oral health maintenance? What was the prognosis of the orthodontic treatment? What about retention?

Response 12: Thank you very much for this comment. Currently, the teeth arch is maintained with the bracket on the labial side. We visit the patient once every month or 2 months to check his teeth arch. We plan retention using a wire on the palatal and lingual side in the future.

Point 13: Are there any further clinical photographs that can be provided, which further show the progress and transition of treatment?

Response 13: Unfortunately, we have no more informative pictures of better quality to add to this revision. We will consider visual data much more carefully in further studies. We apologize for not being able to resolve this issue.

Point 14: A recent publication (Modha, B., 2021. Global Developmental Delay and Its Considerations in Paediatric Dental Care—A Case Report. Oral1(3), pp.181-189) contains similar material to your Introductory material, particularly lines 75-76: “Malalignment of the teeth arches is caused by involuntary and hypertonic facial, masticatory, and tongue muscle movement, and its severity depends on the level of brain damage;” this publication would be a useful one to reference here

Response 14: Thank you for the valuable suggestion. We added this publication to our reference (Page 8, Lines 320).

Point 15: Some other useful and helpful reads and references for your paper: (Abeleira, M.T., Outumuro, M., Diniz, M., García-Caballero, L., Diz, P. and Limeres, J., 2016. Orthodontic treatment in children with cerebral palsy. Cerebral Palsy-Current Steps, pp.130-40), (Napoli, J.A., Drew, S. and Jaeger, T.C., 2020. Management of Skeletal Facial Deformation and Malocclusion in Cerebral Palsy. Cerebral Palsy, pp.1105-1120)

Response 15: Thank you for your suggestion to improve our article. We have added this publication to our references (Page 5, Line 204; Page 8, Lines 340–343).

Point 16: Generally, this paper is quite easy to read. The figures and references are fine. Yet, please kindly note, the language, grammar, punctuation, spelling and sentence structures within the paper, all must be carefully assessed and refined to ensure a succinct and coherent read. This is because there are minor discrepancies in the language, grammar, punctuation, spelling and sentence structures. Please obtain the necessary scientific English language reading and editing assistance, if need be, so that the paper has the potential to be read enjoyably by the international readership.

Response 16: Thank you for this comment. Before this resubmission, an English-speaking academic professional proofread the revised article.

Point 17: Well done for providing this patient with life-changing orthodontic treatment, which I hope has improved their quality of life

Response 17: Thank you very much for this comment. We hope to continue writing articles that share our experience and knowledge with other clinicians and researchers to improve patients’ quality of life.

Reviewer 3 Report

Dear authors, I read your article with pleasure. As is well known, the role of the dentist is of fundamental importance in diagnosing and assessing the degree of OSAS in patients. In fact, through dental care and the use of specific oral devices, patients can have an improvement in this pathology that very often remains undiagnosed.

To increase the value of your article, I recommend making a few corrections:

  • The images are of poor quality: Authors are requested to modify the parameters of the images to achieve better quality.
  • Osas is a condition that very often remains undiagnosed. Therefore, questionnaires such as the STOP-BANG can be a valuable weapon in assessing the risk of OSAS in dental patients. I recommend reading and discussing this article https://doi.org/10.3390/ijerph181910277 .

Author Response

Point 1: Dear authors, I read your article with pleasure. As is well known, the role of the dentist is of fundamental importance in diagnosing and assessing the degree of OSAS in patients. In fact, through dental care and the use of specific oral devices, patients can have an improvement in this pathology that very often remains undiagnosed.

To increase the value of your article, I recommend making a few corrections:

The images are of poor quality: Authors are requested to modify the parameters of the images to achieve better quality.

Response 1: Thank you for your valuable comments that have improved our article. Unfortunately, we have no more informative pictures of better quality to add to this revision. We will consider visual data much more carefully in further studies. We apologize for not being able to respond to this comment.

Point 2: Osas is a condition that very often remains undiagnosed. Therefore, questionnaires such as the STOP-BANG can be a valuable weapon in assessing the risk of OSAS in dental patients. I recommend reading and discussing this article https://doi.org/10.3390/ijerph181910277

Response 2: Thank you very much for your suggestion to improve our article. OSA is sometimes undiagnosed; hence, we consider that dentistry plays an important role in sleep medicine. We have added this publication to our references (Page 8, Lines 328–329).

Round 2

Reviewer 2 Report

Following the authors efforts in making additions and amendments to the manuscript, the manuscript has now reached a standard where publication can be considered. One issue that the authors may need to consider though is the iatrogenic effects of prolonged orthodontic treatment. The patient has had orthodontic treatment for over six years. Could there be adverse effects from this? How will this be monitored? How will this be dealt with? Please consider this.

Once the authors have revisited their paper, thoroughly assessed and polished it to ensure a succinct, professional and coherent read, it would be expected that the paper shall be in good stead for publication. Thus, the paper is currently deemed: Accept after Minor Revision.

Author Response

Point 1: Following the authors efforts in making additions and amendments to the manuscript, the manuscript has now reached a standard where publication can be considered. One issue that the authors may need to consider though is the iatrogenic effects of prolonged orthodontic treatment. The patient has had orthodontic treatment for over six years. Could there be adverse effects from this? How will this be monitored? How will this be dealt with? Please consider this.

Once the authors have revisited their paper, thoroughly assessed and polished it to ensure a succinct, professional and coherent read, it would be expected that the paper shall be in good stead for publication. Thus, the paper is currently deemed: Accept after Minor Revision.

Response 1: Thank you very much for your valuable comment on improving our article. We took six years to complete the orthodontic treatment for the following reasons: (1) weak orthodontic force to avoid pain, and (2) suspension of the treatment because of the patient's hospitalization and the pandemic of COVID-19. The patient may have felt uncomfortable during the treatment because of the fixed multi-bracket appliance. In addition, the patient had a risk of gingivitis and dental caries. In this case, we provided a regular dental checkup and tooth brushing instructions for the mother to reduce the risk of these possible adverse effects. Although the environment of at-home orthodontic treatment has difficulties on patient's posture, field isolation, and the shortage of equipment, we considered that the treatment should finish in a short period. In addition, preventing teeth arch malalignment is important as at-home orthodontic treatment for children with cerebral palsy burdens both dentists and caregivers. We stated them in the main text (Page 6, Lines 235-243).

This manuscript is a resubmission of an earlier submission. The following is a list of the peer review reports and author responses from that submission.